# *In vivo* tracking of transplanted macrophages with near infrared fluorescent dye reveals temporal distribution and specific homing in the liver that can be perturbed by clodronate liposomes

**Satoshi Nishiwaki[1], Shigeki Saito[2], Kyosuke Takeshita[3], Hidefumi Kato[4], Ryuzo Ueda[5], Akiyoshi Takami[6], Tomoki Naoe[7], Mika Ogawa[8], Takayuki Nakayama[8]***

1 Department of Hematology and Oncology, Nagoya University Graduate School of Medicine, Nagoya, Japan, 2 Departments of Hematology, Japanese Red Cross Nagoya Daiini Hospital, Nagoya, Aichi, Japan, 3 Department of Clinical Laboratory, Saitama Medical Center, Kawagoe, Saitama, Japan, 4 Department of Transfusion Medicine, Aichi Medical University, Nagakute, Aichi, Japan, 5 Tumor Immunology, Aichi Medical University, Nagakute, Aichi, Japan, 6 Hematology, Aichi Medical University, Nagakute, Aichi, Japan, 7 Departments of Hematology, Nagoya Medical Center, Nagoya, Aichi, Japan, 8 Clinical Laboratory, Aichi Medical University, Nagakute, Aichi, Japan

* tnaka@aichi-med-u.ac.jp

**Data Availability Statement:** All relevant data are within the manuscript.

## Abstract

Macrophages play an indispensable role in both innate and acquired immunity, while the persistence of activated macrophages can sometimes be harmful to the host, resulting in multi-organ damage. Macrophages develop from monocytes in the circulation. However, little is known about the organ affinity of macrophages in the normal state. Using *in vivo* imaging with XenoLight DiR®, we observed that macrophages showed strong affinity for the liver, spleen and lung, and weak affinity for the gut and bone marrow, but little or no affinity for the kidney and skin. We also found that administered macrophages were still alive 168 hours after injection. On the other hand, treatment with clodronate liposomes, which are readily taken up by macrophages via phagocytosis, strongly reduced the number of macrophages in the liver, spleen and lung.

## Introduction

Macrophages are a type of myeloid cell that play an indispensable role in both innate and acquired immunity [1]. Macrophage phenotypes and functions can vary with different external stimuli, and macrophages are divided into two major classifications: classically activated, i.e. inflammatory, and alternatively activated, i.e. anti-inflammatory macrophages [1]. Persistence of activated macrophages can sometimes be harmful to the host [2, 3]. Hemophagocytic lymphohistiocytosis (HLH), or macrophage activation syndrome (MAS), is a hyperinflammatory state with a high mortality rate and a therapeutic challenge characterized by inappropriate survival of histiocytes and cytotoxic T cells (CTLs), leading to a cytokine storm,

**Funding:** Japanese Grant-in-Aid for Scientific Research (https://www.jsps.go.jp/) Grant number [(C) 17K09944] was awarded to T. Nakayama. The funders had no role in study design, data collection and analysis, decision to publish, or preparation of the manuscript.

**Competing interests:** The authors have declared that no competing interests exist.

hemophagocytosis and multi-organ damage [4]. Monocytes in the circulation, originating from hematopoietic precursors in the bone marrow, lodge in certain organs and eventually develop into tissue macrophages [5]. Thus, the analysis of the mechanism of macrophage homing could lead to new treatment options. However, methods to track macrophages *in vivo* have not been established to date.

Clodronate is a biphosphanate clinically approved for human use as a bone resorption inhibitor and antihypercalcemic agent. Clodronate is metabolized to a toxic adenosine 5'-triphosphanate inside mammalian cell, resulting in cell death by apoptosis [6]. Therefore, efficient delivery of clodronate into the cytoplasm could enhance their therapeutic effect. It is known that clodronate liposomes are readily taken up mainly by macrophages and polymorphonuclear leukocyte via phagocytosis and strongly attenuate the number of macrophages retained in the cytoplasm [7, 8]. Additionally, activated macrophages are more sensitive to clodronate than resting macrophages [8].

Here, we evaluated the organ affinity and fate of transplanted macrophages using *in vivo* imaging with XenoLight DiR®. We also report that treatment with clodronate liposomes efficiently attenuates the number of macrophages in the liver, spleen and lung, for which macrophages possess strong affinities.

## Materials and methods

### Mice

This study was carried out in strict accordance with the recommendations in the Guide for the Care and Use of Laboratory Animals of the National Institutes of Health. The protocol was approved by the Committee on the Ethics of Animal Experiments of the Nagoya University (Protocol Number: 24298). All procedures was performed under sodium pentobarbital anesthesia, and all efforts were made to minimize suffering.

Male 6- to 8-week-old male C57BL/6J mice were obtained from Chubu Kagaku Shizai (Nagoya, Japan). Animals were maintained at constant ambient temperature (22±1°C) under a 12-h light/dark cycle (lights on between 9:00 and 21:00), with food and water available *ad libitum.*

### Preparation of peritoneal macrophages

Thioglycolate-recruited macrophages were isolated from C57BL/6J mice by peritoneal lavage 3 days after intraperitoneal injection of 1 mL 2% sterile thioglycolate medium as described elsewhere with slight modification [9, 10]. Briefly, 20 g of dehydrated Brewer thioglycolate medium powder (Kanto Chemical Co. Inc., Tokyo, Japan) was dissolved in 1000 mL deionized water and autoclaved for 20 minutes at 15 pounds of pressure (121°C). The autoclaved medium was kept in the dark under sterile conditions at room temperature for at least 3 months before use. Obtained peritoneal macrophages were purified using CD11b microbeads and the AutoMACS system (Miltenyi Biotec, Bergisch Gladbach, Germany). The cell viabilities were evaluated by trypan blue exclusion.

### *In vitro* assay for the effects of liposomal clodronate on macrophages

A murine macrophage cell line, RAW264.7, was purchased from American Type Culture Collection (Manassas, VA) and was cultured in Dulbecco's Modified Eagle Medium (DMEM) with 10% fetal bovine serum (FBS). Liposomal clodronate (lipo-CL2MDP) was kindly gifted by Dr N. van Rooijen [11, 12]. Liposomal PBS (lipo-PBS) was used as a control. RAW264.7 cells treated with the indicated concentrations of lipo-CL2MDP or lipo-PBS were seeded at a density of 10,000 cells/well in a 96-well plate and incubated for 48 hours at 37°C. After the

indicated time, the viability of the cells was assessed by a colorimetric assay (TetraColor One®; Seikagaku Co., Tokyo, Japan) as described elsewhere [10, 13, 14]. Briefly, 10 μL of TetraColor One® was added to each well, and the mixture was incubated for an additional 4 hours to measure the viability of cells. Absorbance at 450 nm was monitored and the $IC_{50}$ values of the cells were calculated.

## Labeling protocol

Macrophages were labeled as described previously with slight modifications [15]. The commercially available lipophilic tracer 1,1-dioctadecyl-3,3,3,3-tetramethylindotricarbocyanine iodide (XenoLight DiR®, Caliper, Hopkinton, MA) was used in all experiments. The substance can be solved in ethanol, and its excitation and emission spectra lie in the near-infrared (NIR) (excitation spectrum, 745 nm; emission spectrum, 800 nm). For the following experiments, macrophages ($1\times10^{6}$/mL of medium) were transfected with 2 mL of DiR®-labeling solution (19.7 μmol/L final concentration) for 5 min, washed 3 times with phosphate-buffered saline (PBS), and resuspended in medium. Because of some variability in the labeling intensity between different preparations of primary murine macrophages, cells from the same labeling reaction were used for every set of *in vivo* experiments.

## Imaging protocol

For the assessment of macrophage distribution and localization in mice, animals were studied in a Xenogen IVIS 200 imager (Caliper) under isoflurane anesthesia at different time points, as described elsewhere [16]. Labeled macrophages ($2\times10^{6}$/mouse) were injected via the tail vein, and 200 μL of lipo-CL2MDP or lipo-PBS was administered 3 hours after macrophage injection. Hair of mice was removed with depilatory cream since fluorescence is absorbed by the black hair. Image acquisition times were 1 second. Seven days after drug administration, mice were euthanized. The major organs were harvested to detect the fluorescence signal. The fluorescence signal intensities in different organs were analyzed by Xenogen Living Image® software, Version 3.1 (WaveMetrix, USA).

## Immunohistochemistry

Immunostaining of mouse liver, spleen and lung specimens was performed as previously described with slight modification [10, 17]. Briefly, after fixation with 10% formaldehyde, samples were embedded in paraffin, cut into 3.5 mm sections, mounted on glass slides coated with silane, deparaffinized in xylene and rehydrated through a series of ethanol solutions. The avidin–biotin complex method was used. For antigen retrieval, the sections were autoclaved at 98˚C for 45 minutes in diluted immunosaver® (1:200). Then, specimens were incubated with normal rabbit serum (Dako; diluted to 1:75), and thereafter with a monoclonal antibody for F4/80 (CI:A3-1, Novus, Littleton, CO) (1:100) for 15 minutes using intermittent microwave irradiation [18, 19]. Sections were then incubated with biotin-labeled rabbit anti-mouse IgG serum (diluted to 1:300; Dako) and a streptavidin–biotin detection kit (Ultra Tech HRP kit®, Beckman Coulter, Brea CA) sequentially. Finally, sections were stained with diaminobenzidin solution.

## Statistical analysis

Statistical significance of group differences was evaluated by Student's t-test between two groups using Excel software (Microsoft, Redmond, WA). Statistical differences between groups were considered significant at $P < 0.05$.

## Results

### Kinetics of *in vivo* imaging of macrophages

To label macrophages, we incubated macrophages with lipophilic tracer 1,1-dioctadecyl-3,3,3,3-tetramethylindotricarbocyanine iodide (XenoLight DiR®) for 5 min, washed 3 times with phosphate-buffered saline (PBS), and resuspended in medium. Neither morphological changes nor inviable cells in the macphages treated with XenoLight DiR® were observed compared to intact macrophages, suggesting no harmful impact of XenoLight DiR® on macrophages. Accordingly, fluorescence signal from macrophages was strongly detectable even 7 days after the injection into mice, suggesting that labeled macrophages were still viable *in vivo* (Fig 1A). In control mice with no labeled macrophages, minimal fluorescence signal was observed (Fig 1A). *In vivo* imaging showed that labeled macrophages gathered in the center of the body even 3 hours after the injection and stayed there for 7 days. Quantitative analysis showed that the difference between the intensity of fluorescence signal from mice which received $5 \times 10^6$ labeled macrophages and that from mice with $2 \times 10^6$ labeled macrophages was clearly distinguishable (Fig 1B), suggesting that fluorescence intensity paralleled the number of macrophages. It also showed that fluorescence intensity in the group of $5 \times 10^6$ labeled macrophages decreased only by 29.3% 168 hours after administration (Fig 1B).

### *In vivo* homing patterns of macrophages

To obtain additional anatomic patterns of macrophage homing, mice were euthanized 7 days after the administration of labeled macrophages. The major organs were harvested to detect the fluorescence signal. As shown in Fig 1C, we observed strong signals in the liver, spleen and lung, and weak signals in the gut and bone marrow, but few signals in the kidney or skin. None or minimal signal was detected in the controls (Fig 1C, upper columns).

### Effects of liposomal clodronate on macrophage viability *in vitro* and *in vivo*

The proliferation of mouse macrophage-like RAW264.7 cells after liposomal clodronate (lipo-CL2MDP) treatment (48 hours) was evaluated using a colorimetric assay (Fig 2A). Liposomal PBS (lipo-PBS) was used as a control. Lipo-CL2MDP inhibited the proliferation of RAW264.7 cells in a dose-dependent manner. Lipo-CL2MDP decreased the viability of RAW264.7 cells by 75% at a concentration of 100 μM while lipo-PBS exerted no inhibitory effect on the proliferation of RAW264.7 cells. Next, we tested the effect of lipo-CL2MDP on macrophages *in vivo*. Labeled macrophages ($2 \times 10^6$/mouse) were injected via the tail vein, and 200 μL of lipo-CL2MDP or lipo-PBS was administered 3 hours after macrophage injection (Fig 2B). The difference in signal intensities between the control mice and lipo-CL2MDP-treated mice became apparent only 3 hours after lipo-CL2MDP administration (Fig 2B). Since then, the signal intensity in the mice that received lipo-CL2MDP decreased continuously and reached a plateau at the 24-hour time point (Fig 2B). Seven days following drug administration, mice were euthanized. The major organs were harvested to detect the fluorescence signal. Strong signals were detectable in the liver, spleen, and lung from control mice as observed above, while weak signals and minimal signal were detectable in the liver, spleen, and lung, respectively, from the lipo-CL2MDP-injected mice (Fig 2C). Immunohistochemical staining with F4/80 visualized macrophages (brown cells) scattered within the liver, spleen and lung from control mice (Fig 2D, first row panels). By contrast, rare macrophages were identified in these organs from lipo-CL2MDP-injected mice (Fig 2D, third row panels). However, no histological differences were observed between the liver, spleen and lung from control mice and those from lipo-CL2MDP-injected mice (Fig 2D, second and fourth panels).

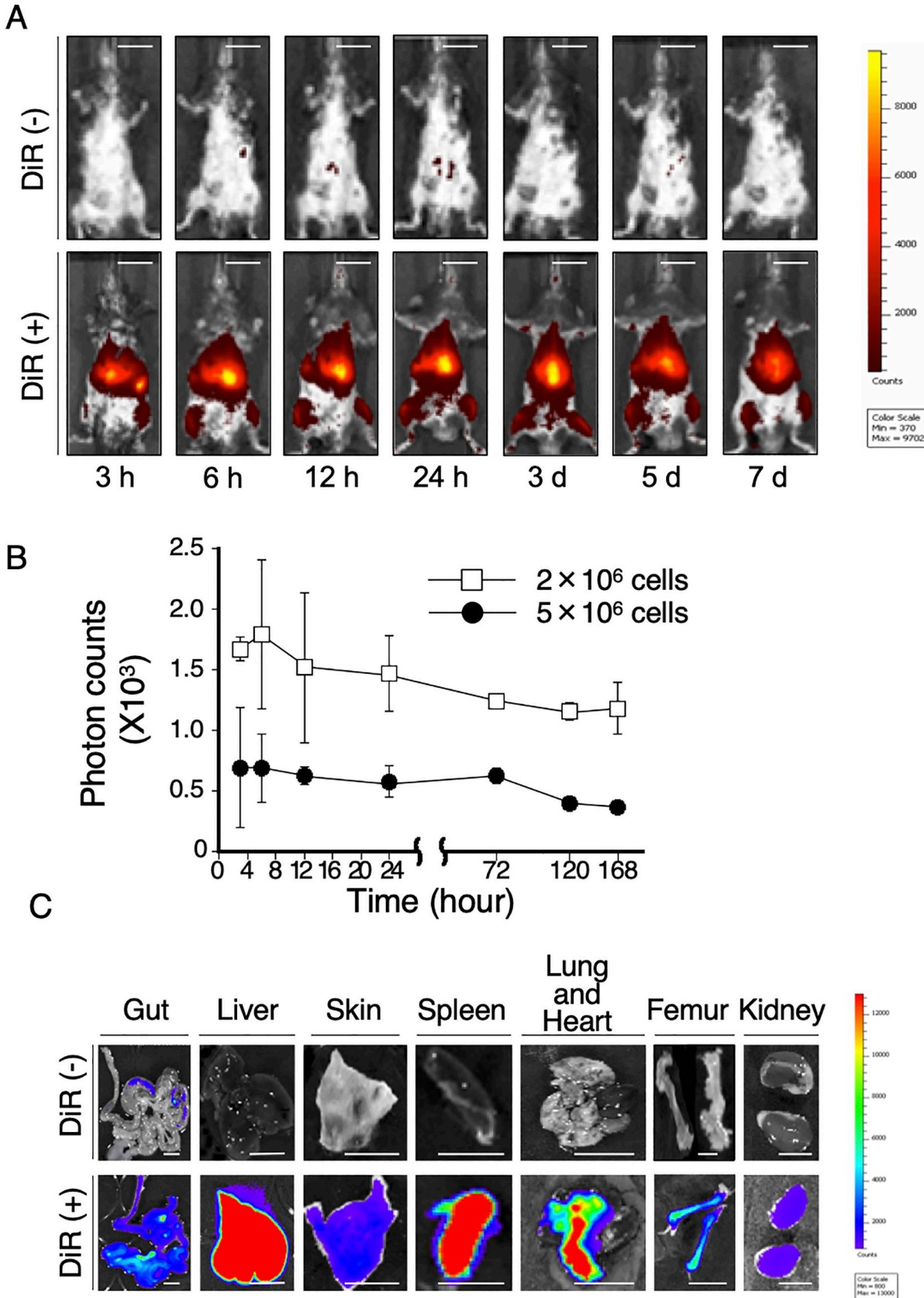

**Fig 1. *In vivo* imaging of macrophages. (A)** Macrophages labeled with/without XenoLight DiR® were injected into mice via tail vein. The time course of total fluorescence signal intensities was evaluated for 7 days. Scale bar, 10 mm. **(B)** Different numbers ($2\times10^6$/body or $5\times10^6$/body) of macrophages labeled with XenoLight DiR® were injected into mice. The total fluorescence signal intensities were monitored for 7 days. **(C)** Seven days after the injection of macrophages labeled with XenoLight DiR®, mice were euthanized. The major organs were harvested to detect the fluorescence signal, showing that labeled macrophages homed specifically in the liver, spleen and lung. Scale bar, 10 mm.

## Discussion

Here we show that XenoLight DiR® is useful for *in vivo* bioluminescent tracking of macrophages. Total fluorescence signal from macrophages decreased only by 29.3% 168 hours after administration (Fig 1A), suggesting that labeled macrophages were still viable *in vivo* and the fluorescence of XenoLight DiR® did not fade for a long period. Similarly, Eisenblätter *et al.* has reported that macrophages labeled with XenoLight DiR® were viable, and successfully tracked when injected intravenously, at the site of inflammation using a cutaneous granuloma mouse model [15].

Previous cell tracking studies showed that most types of cells were initially trapped in the lung, which is enriched with small capillaries, and then migrated to specific organs [20, 21]. Similarly, macrophages were trapped in the lung, but stayed there even 7 days after intravenous injection. Thus, we concluded that Thioglycolate-recruited macrophages possess a specific affinity for the lung, in addition to the liver and spleen (Fig 1C). Thioglycolate-recruited macrophages are activated macrophages expressing immature myeloid markers: Ly6C and ER-MP58 [22]. Their homing patterns are somewhat different those of macrophages established from bone marrow progenitor cells. Eisenblätter *et al.* has reported that macrophages established from bone marrow progenitor cells mainly distributed to the lungs, bone marrow and liver [15]. It is well known that proinflammatory chemokines, such as CCL2, recruit monocytes and macrophages to the inflammatory site [14]. However, the mechanism for the strong affinities of macrophages for these organs has not been fully evaluated to date, even though the lung, liver and spleen include a large number of macrophages. It can be assumed that these organs possess a common mechanism for attracting macrophages.

Haemophagocytic lymphohistiocytosis (HLH), or macrophage activation syndrome (MAS), is a life-threatening condition that causes multi-organ damage, especially liver dysfunction with splenomegaly, and lung involvement [4, 23]. This specific organ damage is of unknown etiology, but can be explained by the high affinities of macrophages for these organs, as described above. A corticosteroid, such as dexamethasone, is a key drug in the treatment of HLH/MAS [24]. However, liposome-encapsulated drugs enhance the anti-macrophage efficacy to a greater extent, compared to intact drugs, because macrophages ingest the drug by phagocytosis [11]. It has been reported that clodronate was converted into adenosine 5'- triphosphate inside cells, which was toxic to macrophages, and then induced cell death by apoptosis [6]. We have reported that dexamethasone palmitate emulsion (DP) treatment clinically ameliorated macrophage-rich graft versus host disease (GVHD) and hemophagocytic syndrome after stem cell transplantation [10, 25]. Here, we showed that macrophages were systemically depleted 24 hours after liposomal clodronate injection (Fig 2C). Similarly, it has been reported that liposomal clodronate efficiently depleted macrophages in large animals, including dogs [26]. As described above, activated macrophages are more sensitive to clodronate than resting macrophages [8]. However, little is known to date about the clinical effects of liposomal clodronate on HLH/MAS. This evidence and results suggest that liposomal clodronate can be applied to treat HLH/MAS.

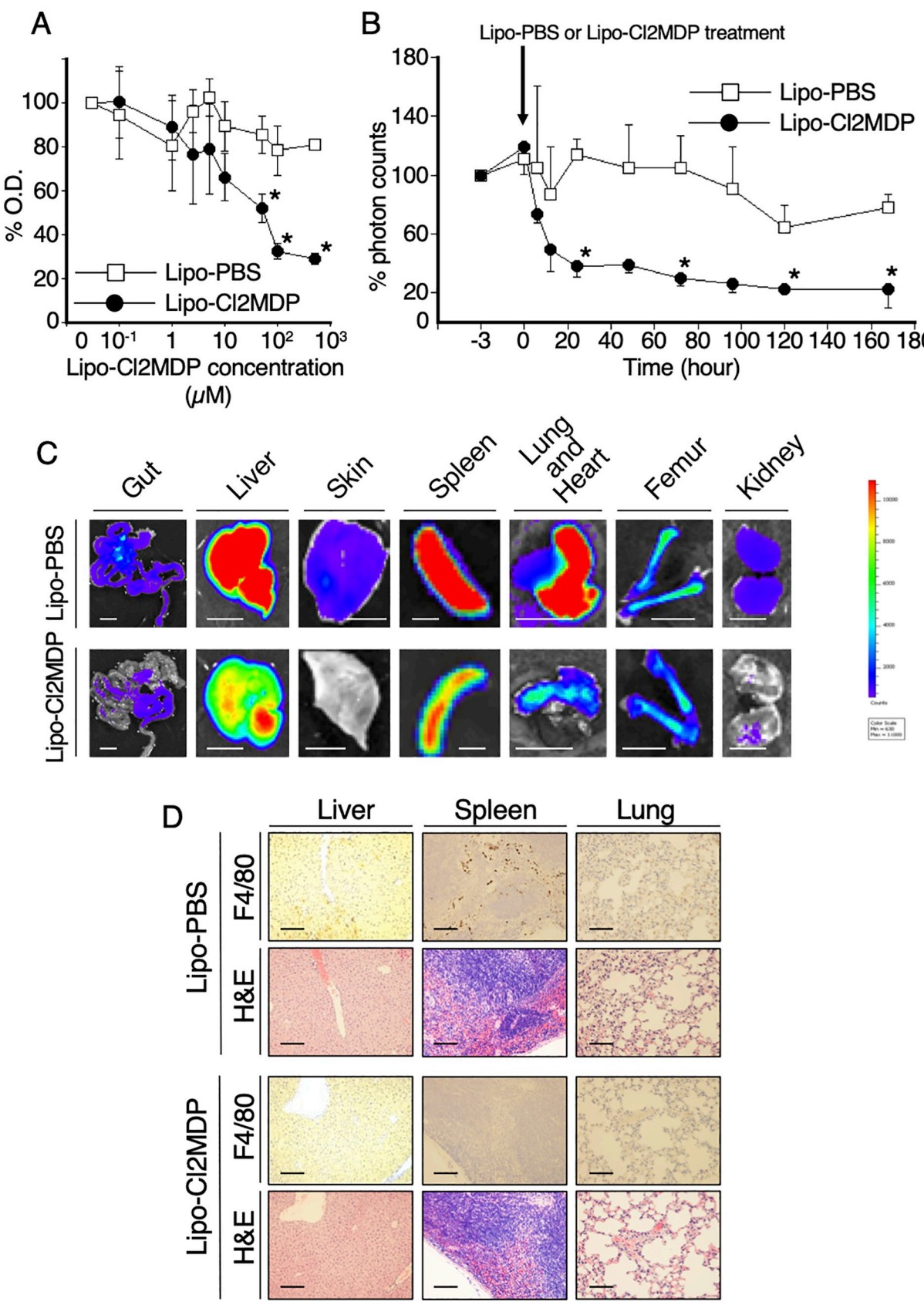

**Fig 2. Effect of liposomal clodronate on macrophages *in vitro* and *in vivo*. (A)** The viability of mouse macrophage-like RAW264.7 cells after liposomal clodronate (lipo-CL2MDP) treatment (48 hours) was evaluated using a colorimetric assay. Liposomal PBS (lipo-PBS) was used as a control. The percentage viability was calculated as follows: (OD value in the presence of each concentration of lipo-CL2MDP or lipo-PBS/OD value with no additive) ×100. The results reflect the mean ± SD of three independent determinations (representative experiment of three performed). The asterisk denotes statistical significance ($^*P < .05$). **(B)** The effects of lipo-CL2MDP on macrophages injected into mice were analyzed by an image analyzer. Lipo-CL2MDP or lipo-PBS was administered into mice 3 hours after the injection of macrophages labeled with XenoLight DiR®. The time course of total fluorescence signal intensities was evaluated for 168 hours. The asterisk denotes statistical significance ($^*P < .05$). **(C)** Lipo-CL2MDP or lipo-PBS was administered into mice 3 hours after the injection of macrophages labeled with XenoLight DiR®. The fluorescence signal intensities in different organs were compared after 7 days. Labeled macrophages home specifically in the liver, spleen and lung (upper panel). However, in the subgroup treated with lipo-CL2MDP (lower panel), fluorescence signal was detected slightly (liver and spleen) and minimally (lung). Scale bar, 10 mm. **(D)** Pathological assessment of macrophages after lipo-CL2MDP treatment. Liver (left panels), spleen (middle panels) and lung (right panels) specimens were stained with H&E (top panels in each subgroup) and the macrophages were detected with the anti-mouse F4/80 monoclonal antibody (bottom panels in each subgroup). Representative images of three mice are shown. Original magnification, ×400. Scale bar, 80 μm.

# Acknowledgments

We are grateful to Ms Rie Goto for her technical help. We thank H. Mizuno for helpful discussions.

# Author Contributions

**Conceptualization:** Satoshi Nishiwaki, Takayuki Nakayama.

**Data curation:** Satoshi Nishiwaki, Hidefumi Kato, Ryuzo Ueda, Akiyoshi Takami, Tomoki Naoe, Mika Ogawa, Takayuki Nakayama.

**Funding acquisition:** Kyosuke Takeshita.

**Investigation:** Satoshi Nishiwaki, Shigeki Saito, Takayuki Nakayama.

**Supervision:** Mika Ogawa, Takayuki Nakayama.

**Validation:** Takayuki Nakayama.

**Writing – original draft:** Takayuki Nakayama.

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
