## [Editor Report · Decision Letter 0]

8 Oct 2019

PONE-D-19-27394

In vivo bioluminescent tracking of transplanted macrophages reveals temporal distribution and specific homing in the liver that can be perturbed by clodronate liposomes

PLOS ONE

Dear Dr. Nakayama,

Thank you for submitting your manuscript to PLOS ONE. Your manuscript requires some relevant revisions before it can be sent out for peer review process. Specifically:

- Figure 1A: Authors need to insert a scalebar in all panels describing the bioluminescence in mice.

- Figure 1C: Authors need to insert a scalebar in all panels depicting the bioluminescence in organs.

- Figure 2C: Authors need to resize all panels depicting the bioluminescence in organs, so that the panels will be displayed with identical sizes.

Figure 2D: Authors must insert a scalebar for each depicted panel in this figure.

Figure legends must be revised in accordance to these new modifications.

We would appreciate receiving your revised manuscript by Nov 22 2019 11:59PM. To enhance the reproducibility of your results, we recommend that if applicable you deposit your laboratory protocols in protocols.io, where a protocol can be assigned its own identifier (DOI) such that it can be cited independently in the future. For instructions see: http://journals.plos.org/plosone/s/submission-guidelines#loc-laboratory-protocols

We look forward to receiving your revised manuscript.

Kind regards,

Fabrizio Mattei, Ph.D.

Academic Editor

PLOS ONE

Journal Requirements:

1. We note that you have included the phrase “data not shown” in your manuscript. Unfortunately, this does not meet our data sharing requirements. PLOS does not permit references to inaccessible data. We require that authors provide all relevant data within the paper, Supporting Information files, or in an acceptable, public repository. Please add a citation to support this phrase or upload the data that corresponds with these findings to a stable repository (such as Figshare or Dryad) and provide and URLs, DOIs, or accession numbers that may be used to access these data. Or, if the data are not a core part of the research being presented in your study, we ask that you remove the phrase that refers to these data.
---

## [Author Response · Author response to Decision Letter 0]

26 Nov 2019

Our responses to each of the comments are as follows. The modifications to the manuscript are tracked in yellow.

- Figure 1A: Authors need to insert a scalebar in all panels describing the bioluminescence in mice.

 We have insert a scalebar in all panels.

- Figure 1C: Authors need to insert a scalebar in all panels depicting the bioluminescence in organs.

 We have insert a scalebar in all panels.

- Figure 2C: Authors need to resize all panels depicting the bioluminescence in organs, so that the panels will be displayed with identical sizes.

 We have made the heights of all panels equal, as the editor suggested.

- Figure 2D: Authors must insert a scalebar for each depicted panel in this figure.

 We have insert a scalebar in all panels.

- Figure legends must be revised in accordance to these new modifications.

 We have mentioned scale bars in the figure legends.

---

## [Decision Letter · Decision Letter 1]

7 Apr 2020

PONE-D-19-27394R1

In vivo bioluminescent tracking of transplanted macrophages reveals temporal distribution and specific homing in the liver that can be perturbed by clodronate liposomes

PLOS ONE

Dear Dr. Nakayama,

Thank you for submitting your manuscript to PLOS ONE. After careful consideration, we feel that it has merit but does not fully meet PLOS ONE’s publication criteria as it currently stands. Therefore, we invite you to submit a revised version of the manuscript that addresses the points raised during the review process.In particular, Authors must carefully and thoroughly address all the points raised by the Reviewer 1. Furthermore, the use of liposomal clodronate is not sufficiently motivated in the text. For this reason, Auhtors must extensively clarify in their manuscript why they studied liposomal clodronate raather than other types of liposomal products.

We would appreciate receiving your revised manuscript by May 22 2020 11:59PM. To enhance the reproducibility of your results, we recommend that if applicable you deposit your laboratory protocols in protocols.io, where a protocol can be assigned its own identifier (DOI) such that it can be cited independently in the future. For instructions see: http://journals.plos.org/plosone/s/submission-guidelines#loc-laboratory-protocols

We look forward to receiving your revised manuscript.

Kind regards,

Fabrizio Mattei, Ph.D.

Academic Editor

PLOS ONE

Reviewers' comments:

Reviewer's Responses to Questions

**Comments to the Author**

1. If the authors have adequately addressed your comments raised in a previous round of review and you feel that this manuscript is now acceptable for publication, you may indicate that here to bypass the “Comments to the Author” section, enter your conflict of interest statement in the “Confidential to Editor” section, and submit your "Accept" recommendation.

Reviewer #1: (No Response)

Reviewer #2: All comments have been addressed

2. Is the manuscript technically sound, and do the data support the conclusions?

Reviewer #1: Partly

Reviewer #2: Yes

3. Has the statistical analysis been performed appropriately and rigorously? 

Reviewer #1: No

Reviewer #2: Yes

4. Have the authors made all data underlying the findings in their manuscript fully available?

Reviewer #1: Yes

Reviewer #2: Yes

5. Is the manuscript presented in an intelligible fashion and written in standard English?

Reviewer #1: No

Reviewer #2: Yes

6. Review Comments to the Author

Reviewer #1: The authors report the localization of fluorescent labelled macrophages within the lungs, spleen and liver and the impact of liposomal clodronate upon maintenance of fluorescence. They demonstrated that labelled macs accumulated flourescence in the above organs, but not kidney or skin. Clodronate is biphosphonate with a liposome and action is not mentioned within the article.

Issues:

1. thioglycollate recruited macrophages are differentially activated through the recruitment process. The old literature (80s) spoke about the type of thiogycollate used and differential ability to phagocytose bacterial. The authors didn't mention the impact/activation of the type of recruitment upon subsequent homing of cells to the various locations. Some mention of cell surface receptors for endothelial or tissue ligands may be useful. Some of the differences are discussed in old literature, again such as J Immunol November 1, 2003, 171 (9) 4816-4823

2. There should be discussion of liposomal clodronate and the induction of apoptosis. It is not mentioned.

3. there is not discussion of statistical analysis of the findings.

4. the authors keep discussing the possibility of liposomal steroids, but did not study it. Liposomal clodronate has a putative role in autoimmune and transplant related conditions where on may want to remove tissue macrophage damage. It was not mentioned within the context of this manuscript. the fact that inert liposomes did not appear to alter macrophages and clodronate loaded liposomes was associated with loss of macrophage associated fluorescence suggests that this may be a way to gain targeted drug delivery, but it is very round about.

Reviewer #2: Nakayama et al presents a new method to track macrophages in vivo using bioluminescence and probe the method of depletion using clodronate liposomes.

Actually the article checks a macrophage tracking method in vivo. Regarding the methodology used, the necessary controls are considered to validate the technique and the materials and methods are adequate to verify that the method works correctly. Perhaps it would be interesting to test it on immunodeficient mice or another animal species to see the possibilities of the technique. The Editor should assess whether it is of sufficient scientific quality to be published on Plos One.

7. PLOS authors have the option to publish the peer review history of their article (what does this mean?). If published, this will include your full peer review and any attached files.

Reviewer #1: No

Reviewer #2: Yes: Carlos Alfaro

---

## [Author Response · Author response to Decision Letter 1]

21 May 2020

response to reviewers

Academic Editor: The use of liposomal clodronate is not sufficiently motivated in the text. For this reason, Authors must extensively clarify in their manuscript why they studied liposomal clodronate rather than other types of liposomal products.

As the academic editor suggested, we have added descriptions about advantages of liposomal clodronate to deplete macrophages (page 3, line56-57 and page 4, line 62-63).

Reviewer #1: The authors report the localization of fluorescent labelled macrophages within the lungs, spleen and liver and the impact of liposomal clodronate upon maintenance of fluorescence. They demonstrated that labelled macs accumulated fluorescence in the above organs, but not kidney or skin. Clodronate is bisphosphonate with a liposome and action is not mentioned within the article.

We are grateful for the useful comments and suggestions that have helped us to improve our paper. We have taken all these comments and suggestions into account in the revised version of our manuscript. 

1. thioglycolate recruited macrophages are differentially activated through the recruitment process. The old literature (80s) spoke about the type of thioglycolate used and differential ability to phagocytose bacterial. The authors didn't mention the impact/activation of the type of recruitment upon subsequent homing of cells to the various locations. Some mention of cell surface receptors for endothelial or tissue ligands may be useful. Some of the differences are discussed in old literature, again such as J Immunol November 1, 2003, 171 (9) 4816-4823

As the reviewer suggested, we have added the description about difference of homing patterns between thioglycolate-recruited macrophages and macrophages established by other means (page 14, line 232-234 and page 15, line 247).

2. There should be discussion of liposomal clodronate and the induction of apoptosis. It is not mentioned.

As the reviewer suggested, we have added the description about apoptotic cell death of macrophages induced by clodronate (page 14, line 232-234 and page 15, line 247). 

3. there is not discussion of statistical analysis of the findings.

As the reviewer suggested, we have added descriptions and asteriks which denotes statistical significance in the Material and Method and Results sections, and in the Figure 2, respectively.

4. the authors keep discussing the possibility of liposomal steroids, but did not study it. Liposomal clodronate has a putative role in autoimmune and transplant related conditions where on may want to remove tissue macrophage damage. It was not mentioned within the context of this manuscript. the fact that inert liposomes did not appear to alter macrophages and clodronate loaded liposomes was associated with loss of macrophage associated fluorescence suggests that this may be a way to gain targeted drug delivery, but it is very round about.

We are very sorry for the errors we made in the page 15, line 238 . Liposomal steroid should have been liposomal clodronate. We have corrected the errors. Additionally, we have changed the description about clodronate (page 3, line 56-58) since it was confusing and roundabout as the reviewer pointed out.

Reviewer #2: Nakayama et al presents a new method to track macrophages in vivo using bioluminescence and probe the method of depletion using clodronate liposomes.

Actually, the article checks a macrophage tracking method in vivo. Regarding the methodology used, the necessary controls are considered to validate the technique and the materials and methods are adequate to verify that the method works correctly. Perhaps it would be interesting to test it on immunodeficient mice or another animal species to see the possibilities of the technique. The Editor should assess whether it is of sufficient scientific quality to be published on Plos One.

We thank the reviewer for helpful suggestions to increase the quality of our manuscript. 

The reviewer pointed out that we should have used appropriate controls in the tracking experiments. It is very difficult to pick up "appropriate controls". However, we have tracked down the fluorescence -labeled mesenchymal stromal cells in vivo after intravenous injection, showing that mesenchymal stromal cells transiently lodged in the lung and then moved to liver and spleen (now preparing for submission). The in vivo behavior of macrophages apparently differs from that of mesenchymal stromal cells. Thus, we concluded that the strong affinities for the lung, liver and spleen were specific to macrophages. 

The reviewer also proposed that we should test it on immunodeficient mice or another animal species to see the possibilities of the technique. We had similar ideas as well, but we gave up to perform experiments since chemokine systems are different between mice and humans (e.g. mouse CCL2 is 58% identical to human CCL2). Thus, we were afraid that we might not obtain accurate data when human cells were injected into immunodeficient mice. Additionally, we don’t have an image analyzer for large animals

---

## [Decision Letter · Decision Letter 2]

1 Oct 2020

PONE-D-19-27394R2

In vivo bioluminescent tracking of transplanted macrophages reveals temporal distribution and specific homing in the liver that can be perturbed by clodronate liposomes

PLOS ONE

Dear Dr. Nakayama,

Thank you for submitting your manuscript to PLOS ONE. After careful consideration, we feel that it has merit but does not fully meet PLOS ONE’s publication criteria as it currently stands. Therefore, we invite you to submit a revised version of the manuscript that addresses the points raised during the review process.

As you can see from the comments, the reviewers felt that the scientific soundness of this study should be improved before the acceptance of this work. One reviewer raised the important issue on quantitative analysis, which needs to be addressed. To help me expedite processing, please explicitly address the questions raised by the reviewers in your cover letter and also point out the changes made in the manuscript. I will go back to the reviewers for further input and advice before any final decision on possible publication is made.

We look forward to receiving your revised manuscript.

Kind regards,

Bing Xu, PhD

Academic Editor

PLOS ONE

Reviewers' comments:

Reviewer's Responses to Questions

**Comments to the Author**

1. If the authors have adequately addressed your comments raised in a previous round of review and you feel that this manuscript is now acceptable for publication, you may indicate that here to bypass the “Comments to the Author” section, enter your conflict of interest statement in the “Confidential to Editor” section, and submit your "Accept" recommendation.

Reviewer #2: All comments have been addressed

Reviewer #3: (No Response)

Reviewer #4: All comments have been addressed

2. Is the manuscript technically sound, and do the data support the conclusions?

Reviewer #2: Yes

Reviewer #3: No

Reviewer #4: Yes

3. Has the statistical analysis been performed appropriately and rigorously? 

Reviewer #2: Yes

Reviewer #3: No

Reviewer #4: Yes

4. Have the authors made all data underlying the findings in their manuscript fully available?

Reviewer #2: Yes

Reviewer #3: Yes

Reviewer #4: Yes

5. Is the manuscript presented in an intelligible fashion and written in standard English?

Reviewer #2: Yes

Reviewer #3: Yes

Reviewer #4: Yes

6. Review Comments to the Author

Reviewer #2: The authors have made an effort to answer the questions raised by the reviewers and are generally acceptable and correct. The article has improved significantly with the changes introduced.

Reviewer #3: The paper by Nakayama and colleagues aim to image the biodistribution of Raw macrophages and see the effect of Clodronate liposomes on their depletion using optical imaging. Since the authors use the XenolighDIR fluorescent dye they are imaging near infrared fluorescence and not bioluminescence so the title is misleading.

Moreover since the perform direct labelling and the dye bound to cell membranes, there is no proof that cells retain it after 24h so for their analysis at 168h the authors should provide more evidence that they are looking at macrophages in the different organs and not at the free dye. Moreover the paper lacks quantitative analysis. The author should report the fluorescence signal in the organ in relation to weight of the organ so that a comparison of the prevalence of the cells can be claimed and statistical analysis performed.

Finally A. Taylor and colleagues ( doi: 10.1155/2018/2514796) injected bioluminescent RAW macrophages in animals and, although they also see signals in the liver , they were able to see it in the brain area too. So it would be nice to add this organ to the analysis.

I am afraid that in this form the paper is more appropriate as a technical note and not a research article.

Reviewer #4: In this manuscript, the authors studied the biodistribution of implanted macrophages in mice model. The results revealed that the macrophages exhibited organ-specific accumulation. This result provides new knowledge for immunologist. Moreover, the authors developed a method using clodronate liposomes to remove the macrophages accumulated in the liver, spleen and lung. This way could potentially address diseases like Haemophagocytic lymphohistiocytosis (HLH), or macrophage activation syndrome (MAS) which are relevant to the high affinities of macrophages. This work is technically sound. The experiment groups and the controls are well designed. The authors have fully address the concerns raised by previous reviewers. Thus, I recommend the acceptance of this manuscript.

7. PLOS authors have the option to publish the peer review history of their article (what does this mean?). If published, this will include your full peer review and any attached files.

Reviewer #2: **Yes: **Carlos Alfaro

Reviewer #3: No

Reviewer #4: No

---

## [Author Response · Author response to Decision Letter 2]

27 Oct 2020

Our responses to each of the comments are as follows. The modifications to the manuscript are tracked in yellow.

Reviewer #3: The paper by Nakayama and colleagues aim to image the biodistribution of Raw macrophages and see the effect of Clodronate liposomes on their depletion using optical imaging. 

Query: Since the authors use the XenolighDIR fluorescent dye they are imaging near infrared fluorescence and not bioluminescence so the title is misleading.

Answer: As pointed out by the Reviewer 3, the term “bioluminescence” was inappropriate. Thus, we changed the title as “In vivo tracking of transplanted macrophages with near infrared fluorescent dye reveals temporal distribution and specific homing in the liver that can be perturbed by clodronate liposomes”. 

Query: Moreover since the perform direct labelling and the dye bound to cell membranes, there is no proof that cells retain it after 24h so for their analysis at 168h the authors should provide more evidence that they are looking at macrophages in the different organs and not at the free dye. 

Answer: The product sheet of Xenolight DIR tells that the two long 18-carbon chains insert into the cell membrane, resulting in specific and stable cell staining with negligible dye transfer between cells (https://www.perkinelmer.com/product/dir-125964). 

There was virtually no free dye in the cell suspension, because we washed macrophages labeled with Xenolight DIR three times with PBS before we injected as reported previously [J Nucl Med 2009; 50: 1676-1682]. It has been reported that primary macrophages retained the Xenoligh DIR fluorescent dye over 7 days [J Nucl Med 2009; 50: 1676-1682].

In Fig2, fluorescence signal intensities were much decreased in the liver, spleen and lung of mice treated with Lipo-CL2MDP, where the number of macrophages was much decreased as well, suggesting that Xenolight DIR inserted into the cell membrane of macrophages were not trasferred to other cells.

We had wanted to identify cells with fluorescence as macrophages, but it was practically impossible because (1) the resolution of in vivo imaging system at our facility was too low to detect a single cell and (2) microscopic images of organs, immunostained with F4/80, could not be overlaid with fluorescence images obtained from in vivo imaging. 

Query: Moreover the paper lacks quantitative analysis. The author should report the fluorescence signal in the organ in relation to weight of the organ so that a comparison of the prevalence of the cells can be claimed and statistical analysis performed.

Answer: 

The intensities of fluorescence signal were expressed as photons/sec/unit area, not total photons of each organ. Thus, thickness, not weight, of organs may correlates with signal intensities. However, there is little and ignorable difference of thickness among organs, except for femors. 

Each organ has unique structure (e.g. bones in the femurs, alveolars in the lung, and blood in the heart), which can attenuate the signal intensities to some extent. Statisitical analysis can not be performed among subgroups in different backgrounds. Thus, these were reasons why we analyzed trends, not statistical significances, in the macrophage accumulation as shown in Figure 1C and 2C. On the other hand, we performed statistical analysis between mice treated with clodronate liposomes and mice treated without clodronate liposomes to confirm the effect of clodronate liposomes on the macrophage accumulation since both subgroups had similar backgrounds.

Query: Finally A. Taylor and colleagues ( doi: 10.1155/2018/2514796) injected bioluminescent RAW macrophages in animals and, although they also see signals in the liver , they were able to see it in the brain area too. So it would be nice to add this organ to the analysis.

Answer: We did not analyze the intensity of fluorescence signal in the brain because minimal signal intensity was observed in the head region as shown Figure 1A after primary macrophages were administrated via tail vein. In the Contrast Media & Molecular Imaging paper (doi.org/10.1155/2018/2514796), the authors injected the luciferase gene-transduced RAW 264.7 cells via the left cardiac ventricle, suggesting that RAW 264.7 cells were directly delivered to the brain via large arteries. RAW 264.7 is a macrophage-like leukemia cell transformed by virus (https://www.atcc.org/products/all/tib-71.aspx). Thus, we believe that the route of administration and the cell type are likely cause of failure for us to detect the signal in the head.

We hope that the Editor finds that our re-revised manuscript adequately addresses the comments made by the reviewer #3 and that the manuscript is now ready for publication in the PLOS One.

---

## [Editor Report · Decision Letter 3]

4 Nov 2020

In vivo tracking of transplanted macrophages with near infrared fluorescent dye reveals temporal distribution and specific homing in the liver that can be perturbed by clodronate liposomes

PONE-D-19-27394R3

Dear Dr. Nakayama,

We’re pleased to inform you that your manuscript has been judged scientifically suitable for publication and will be formally accepted for publication once it meets all outstanding technical requirements.

Kind regards,

Bing Xu, PhD

Academic Editor

PLOS ONE
---

## [Editor Report · Acceptance letter]

24 Nov 2020

PONE-D-19-27394R3 

*In vivo* tracking of transplanted macrophages with near infrared fluorescent dye reveals temporal distribution and specific homing in the liver that can be perturbed by clodronate liposomes 

Dear Dr. Nakayama:

I'm pleased to inform you that your manuscript has been deemed suitable for publication in PLOS ONE. Congratulations! Your manuscript is now with our production department. 

Kind regards, 

on behalf of

Dr. Bing Xu 

Academic Editor

PLOS ONE